# Licorice Extract Supplementation Benefits Growth Performance, Blood Biochemistry and Hormones, Immune Antioxidant Status, Hindgut Fecal Microbial Community, and Metabolism in Beef Cattle

**DOI:** 10.3390/vetsci11080356

**Published:** 2024-08-06

**Authors:** Sunzhen Liang, Jinzhu Meng, Zining Tang, Xinxin Xie, Miaomiao Tian, Xiaowan Ma, Xiao Yang, Dingfu Xiao, Shuilian Wang

**Affiliations:** 1College of Animal Medicine, Hunan Agricultural University, Changsha 410128, China; sunzhenliang2024@stu.hunau.edu.cn (S.L.); mjz122021@stu.hunau.edu.cn (J.M.); tangzininga@163.com (Z.T.); xinxinxie316@163.com (X.X.); tianmiaomiao815@163.com (M.T.); mxw9921@163.com (X.M.); yx136244359@163.com (X.Y.); 2College of Animal Science and Technology, Hunan Agricultural University, Changsha 410128, China

**Keywords:** licorice extract, growth performance, fecal microbiota, serum parameter, fecal metabolome, beef cattle

## Abstract

**Simple Summary:**

The long-term use of antibiotics has led to problems such as resistance, with serious implications for global public health. Herbs are considered ideal for new antibiotic alternatives because they are natural, non-resistant, non-toxic, or low in toxicity. Licorice extract has been shown to benefit growth performance and health when added to the diets of pigs, chickens, and sheep. However, the use of licorice extract in beef cattle farming is still relatively rare. Therefore, this study aimed to investigate the effects of licorice extract on growth performance, blood index, hindgut microbiome, and metabolome of beef cattle. The results demonstrated that adding licorice extract to the beef cattle diet played a positive role in enhancing the growth performance, blood indexes, and intestinal health of beef cattle.

**Abstract:**

This study aimed to evaluate the effects of licorice extract (LE) on growth performance, nutrient apparent digestibility, serum index (biochemistry, hormones, humoral immunity, and antioxidant function), hindgut fecal microbiota, and metabolism in beef cattle. In total, 12 male yellow cattle aged 12 months were divided into two groups (6 cattle per group): the basal diet (CK group) and the basal diet supplemented with 2 g/kg LE (CHM group). The entire experimental phase lasted for 120 days, including a 30-day pre-feeding period. Compared to the CK group, the average daily gain, crude fiber, calcium, and crude protein nutrient digestibility were greater on d 30 than d 60 (*p* < 0.05) and the feed meat ratio was lower for LE addition (*p* < 0.01). In terms of serum indexes, the insulin and nitric oxide contents were enhanced on d 30, the alkaline phosphatase level was improved on d 60, and the levels of albumin, immunoglobulin A, and catalase were increased on d 90 (*p* < 0.05). In contrast, the cholesterol content was lower on d 60 for LE addition compared with the CK group (*p* < 0.05). The higher enrichment of [Eubacterium]-oxidoreducens-group, p-2534-18b5-gut-group, and Ileibacterium were observed in the CHM group (*p* < 0.05), while the relative abundances of Gallibacterium and Breznakia in the CHM group were lower compared with the CK group (*p* < 0.05). In addition, the differential metabolites related to healthy growth in the CHM group were increased compared with the CK group. And there was a close correlation between hindgut microbiota and metabolic differentials. In general, LE has a promoting effect on the growth performance and health status of beef cattle over a period (30 to 60 days).

## 1. Introduction

Over the past few decades, antibiotics have been added to animal diets as growth promoters. Despite various limitations due to the residues and resistance of antibiotics, a good deal of antibiotics are still used globally every year [1,2]. Therefore, it is urgent that a safe and environmentally friendly additive is established to replace antibiotics. LE, a plant-based additive, was reported to have a variety of nutritional effects, which can improve growth and have pharmacological effects including immune regulation, antioxidant, and anti-inflammation in various animals [3]. Naseri Moghadam et al. found that LE could significantly increase the feed-to-meat (F/G) and feed utilization efficiency in fattened lambs [4]. It is reported that supplementing LE in diets enhanced the growth performance and the antioxidant status in broilers [5]. In addition, glycyrrhiza polysaccharides (a main active component of LE) addition improved the average daily gain (ADG) and average daily feed intake (ADFI), reduced the F/G, and increased intestinal beneficial bacteria so as to benefit the growth and intestinal health of weaned piglets [6]. Zhang et al. reported that glycyrrhiza polysaccharide dietary supplement promoted growth performance, improved serum biochemistry, and enhanced the antioxidant capacity in broilers [7]. Hence, supplementation of LE might had a greater increase in growth performance in beef cattle.

However, there were few studies relevant to LE on finishing beef cattle’ growth and health status, which was likely to provide a new way to replace antibiotics, hormones, and agricultural chemical drugs to improve the healthy growth of beef cattle. We hypothesized that dietary supplementation of LE could enhance the growth and health status of beef cattle. Therefore, the aim of this study was to investigate the effect of LE on growth, apparent digestibility of nutrients, blood biochemistry, blood hormones, immune performance, antioxidant function, hindgut fecal microbiota, and metabolome in beef cattle. The addition of LE in the diet of beef cattle was also conducive to the maintenance of food safety and the healthy development of the livestock industry, which will also provide a theoretical basis for its application in beef cattle.

## 2. Materials and Methods

### 2.1. Animal Ethics

The animal care and treatment procedures were approved by the Animal Ethics Committee of Hunan Agricultural University (No. 432052331).

### 2.2. Animal Management and Experimental Design

All cattle used in this experiment were raised in Ruilong Beef Cattle Co., Ltd., (Changsha, China). Twelve male yellow beef cattle aged 12 months were used based on the principle of similar weight (210 ± 30 kg) and randomly divided into two groups, consisting of a control group (CK; supplementation with regular basal diet, *n* = 6) and an experimental group (CHM; regular basal diet supplemented with 2 g/kg LE, *n* = 6). The trial period was 120 days, consisting of a 30-day pre-trial period and a 90-day main trial period (data and sample collection). The LE used in this study was purchased from Shaanxi Xinnuote Biotechnology Co., Ltd., Xi’an, Shaanxi Province, China. (No. 97676-23-8, containing 50% polysaccharide) and added to the total mixed ration (TMR) diet under the same conditions. The specific nutritional components are shown in Appendix A. All beef cattle were individually housed (3 m × 3 m) and fed at 9:00 and 16:00 daily, allowing them to feed and drink freely while recording the amount of feeding and monitoring their health at all times.

### 2.3. Determination of Active Components in LE

The metabolites in LE were detected by liquid chromatography-tandem secondary mass spectrometry (LC-MS/MS). In total, 1736 metabolites were identified in LE. The top 10 bioactive constituents included sugars and their derivatives (50.46%), flavonoids (11.86%), organic acids and their derivatives (6.32%), amino acids and their derivatives (5.31%), organic heterocyclic compounds (5.27%), alkaloids and their derivatives (4.68%), lipids (4.39%), terpenes (3.14%), phenylpropanoids (2.53%), and phenolic acids (2.04%).

### 2.4. Sample Collection and Analysis

#### 2.4.1. Performance of Growth

During the experiment, the daily intake and residual feed of beef cattle were recorded and some feed samples were taken to determine the moisture content and dry matter content, thereby calculating the dry matter intake of each group of beef cattle. At the beginning of the trial and the end of each month, beef cattle were weighed before the morning feed to calculate the ADG. Dry matter intake was recorded daily to calculate ADFI and F/G.

#### 2.4.2. Collection of Feed and Fecal Samples and the Analysis of Apparent Feed Digestibility

TMR samples were collected on d 0, 30, and 60 of the formal experiment and mixed evenly using the quartering method, followed by drying under 65 °C to determine the conventional nutritional components and acid-insoluble ash contents in the feed.

On the 30th, 60th, and 90th day of the formal experiment, 200 g of manure from beef cattle was collected by a quartering method (1000 g of beef cattle feces were collected in the morning and evening for three consecutive days and then mixed all together). After drying under 65 °C, the feces were placed at room temperature for 24 h to regain moisture. Then, the feces samples were dried again and screened through a 40-mesh filter to determine the conventional nutrients and acid-insoluble ash in the feces. Crude protein (CP), crude fiber (CF), crude fat (EE), crude Ash (Ash), Calcium (Ca), Phosphorus (P), and acid insoluble ash (AIA) were detected according to China National Standards [8,9,10,11,12,13,14]. The measurement methods of neutral detergent fiber (NDF) and acid detergent fiber (ADF) were according to Van Soest et al. [15]. Total energy (GE) was measured using a Bomb Calorimeter. The apparent digestibility of nutrients (ADN) was calculated as follows:$$ADN\left(\%\right)=\left(1-\frac{AIA\;in\;diet\;(\%)}{AIA\;in\;digesta\;(\%)}\times \frac{Digesta\;nutrient\;(\%)}{Diet\;nutrient\;(\%)}\right)\times 100$$

#### 2.4.3. Collection and Analysis of Blood

On the 30th, 60th, and 90th day of the formal experiment, all cattle were subjected to caudal vein blood sampling before the morning feed following a centrifuge at 3500 r/min for 15 min. The serum was separated and divided into 1.5 mL centrifuge tubes, labeled, and placed at −20 °C.

Blood biochemical indexes (AST, ALP, TP, ALB, UREA, TG, LDL-C, HDL-C, CHOL, and ALT), immune performance indexes (IgA, IgG, and IgM), aspartate aminotransferase (AST) and alkaline phosphatase (ALP), total protein (TP), albumin (ALB), urea (UREA), creatinine (CREA), triglyceride (TG), low-density lipid protein (LDL-C), high-density lipoprotein (HDL-C), cholesterol (CHOL), and alanine aminotransferase (ALT) were measured by an Inovo automatic biochemical analyzer. Blood hormone indicators (insulin (INS) and growth hormone (GH)) were measured by the production kit of Beijing North Institute of Biotechnology Co., Ltd., Beijing, China. Blood antioxidant indexes (catalase (CAT), glutathione peroxidase (GSH-Px), total antioxidant energy (T-AOC), nitric oxide (NO), malondialdehyde (MDA), and total superoxide dismutase (SOD)) were measured by kits.

#### 2.4.4. Analysis of the Fecal Microbiota in the Hindgut

On the 91st day, the feces were collected from the rectum of 12 beef cattle and stored separately at −80 °C. The total DNA from the samples was extracted using a Mag-Bind Soil DNA Kit (Omega Bio-Tek, Norcross, GA, USA) and DNA purity was determined by Agarose gel electrophoresis, while its content was analyzed using a NanoDrop 2000 spectrophotometer (Thermo Scientific, Waltham, MA, USA). The 16S V3+V4 region primers (341f and 806r) were selected for sequencing to analyze the microbial diversity. PE libraries were constructed using the TruSeq™ DNA Sample Prep Kit (Illumina, San Diego, CA, USA) and sequenced using the Miseq PE300 platform. Microbial community analysis was performed in the bioscience cloud (https://www.bioincloud.tech, accessed on 28 July 2023).

#### 2.4.5. Fecal Untargeted Metabolome Analysis

Fecal samples (100 mg) were blended with 500 μL of 80% methanol aqueous solution standing in an ice bath for 5 min before 20-min centrifugation (4 °C and 15,000 rpm) to obtain supernatants for LC-MS analysis via an ACQUITY UPLC system coupled with a Waters Xevo G2-XS Q-TOF/MS spectrometer (Waters, Milford, MA, USA). A waters ACQUITY UPLC BEH C18 column (2.1 × 50 mm, 1.7 μm) was used for metabolite separation. The blank sample was replaced with 53% methanol. Metabolome analysis was performed in the bioscience cloud (https://www.bioincloud.tech, accessed on 28 July 2023).

### 2.5. Data Analysis

SPSS 23.0 statistical software was used to analyze the data by ANOVA with a subsequent Student’s *t*-test and a significant difference was inferred for *p* < 0.05 and an extremely significant difference was inferred for *p* < 0.01. Correlation analysis of different metabolites and different bacteria genera was performed in the bioscience cloud (https://www.bioincloud.tech, accessed on 28 July 2023).

## 3. Results

### 3.1. Performance of Growth and Apparent Nutrient Digestibility

On d 30 to 60, the ADG was improved in the CHM group (*p* < 0.01; Table 1) and the F/G ratio declined compared with the CK group (*p* < 0.01). Beef cattle supplied with LE had a higher Ca and CF digestibility (*p* < 0.05) on d 30 and greater CP digestibility (*p* < 0.05) on d 60 compared with the CK group.

### 3.2. Serum Biochemistry and Hormone Parameters

On d 60, the CHOL content was decreased and the ALP concentration was increased in the CHM group compared with that in the CK group (*p* < 0.05; Table 2). On d 90, the ALB content in the CHM group was higher than that in the CK group (*p* < 0.05). On d 30, LE supplementation in beef cattle diets resulted in a higher INS content than the CK group (*p* < 0.05).

### 3.3. Blood Immune Parameters

As shown in Table 3, on d 90, the serum IgA concentration of the CHM group was higher than that of the CK group (*p* < 0.05).

### 3.4. Blood Antioxidant Parameters

The CAT activity of the CHM group was higher on d 90 and the NO levels of CHM exhibited higher values on d 30 than that of the CK group (*p* < 0.05; Table 4).

### 3.5. The Hindgut Fecal Microbiota

The v3-v4 regions of 16s rRNA from fecal samples of 12 cattle were sequenced by the Illumina NovaSeq sequencing platform. In total, 446,054 sequences were obtained from 12 fecal samples. After OTU clustering of sequences, 6227 OTUs were obtained. The number of OTUs shared by the CK and the CHM groups was 1667. In total, 2269 and 2291 unique OTUs were obtained from the CK and the CHM groups, respectively (Figure 1).

As indicated in Figure 2, the CHM group overlaps with the CK group through the beta diversity analysis of beef cattle fecal samples in each group by PCoA. Axis 1 and Axis 2 represented 18.4% and 12.3%, respectively.

The Alpha Diversity Analysis of hindgut fecal microbiota showed no significant difference among groups found in various parameters (*p* > 0.05; Appendix A).

As shown in Figure 3, the dominant colonies at the phylum level were Firmicutes and Bacteroidota. The abundance of Ileibacterium, p-2534-18b5-gut-group, and [Eubacterium]-oxidoreducens-group were greater (*p* < 0.05) and the abundance of Gallibacterium and Breznakia were lower (*p* < 0.05), compared with the CK group (Table 5).

### 3.6. Hindgut Fecal Metabolome

Quality control (QC) samples in both cationic and anionic modes were relatively tightly clustered, indicating that the LC-MS method had good reproducibility and stability (Appendix A).

The metabolic spectrum analysis was performed to plot the metabolic products after being supplied with LE and the metabolites of the intestinal feces were separated and clustered in different regions (Figure 4).

In total, 39 differential metabolites were obtained between the two groups (Figure 5) and 14 metabolites were identified in fecal samples based on KEGG analysis. Corticosterone, cotodorpine, and erucic acid were lipids and lipid-like molecules; Biotin and stercobilin were organic heterocyclic compounds; 2-phenylethylamine and styrene were benzenoids (Table 6).

The selected differential metabolites were concentrated in three different metabolic pathways including steroid hormone biosynthesis, phenylalanine metabolism, and biotin metabolism (*p* < 0.05; Table 7).

The correlation analysis between the different bacteria genera and the hindgut fecal metabolome differentials is shown in Figure 6. Positive correlations between Gallibacterium and maltotriose (*p* < 0.01) and styrene, 2-phenylethylamine, and 3-methylindole (*p* < 0.05) and a negative correlation with corticosterone (*p* < 0.01) and cortodoxone (*p* < 0.05) were found. Ileibacterium was negatively correlated with AICAR (*p* < 0.05). There was a positive correlation between P-2534-18B5-group and eicosapentaenoic acid (*p* < 0.01), 4-hydroxyretinoic acid, corticosterone, and cortodoxone (*p* < 0.05), and D-threo-Isocitric acid (*p* < 0.01) but a negative correlation with 3-methylindole and AICAR (*p* < 0.05). A positive correlation between Breznakia and styrene was found (*p* < 0.01), which was negatively correlated with 5-hydroxyindole-3-acetic acid (*p* < 0.01).

## 4. Discussion

Previous studies have demonstrated that supplementing LE in diets could improve the growth performance of lambs [4] and fish [16], the immune antioxidant status of chickens [17], calves [18], and pigs [19], the embryo production performance of cows [20], and maintain the intestinal health of chickens [21]. Conversely, LE has been less studied for the growth and health status of finishing beef cattle. Our study discovered that exploring LE as a dietary additive could improve the growth performance and health of beef cattle (30 to 60 days), providing data for the application of LE in beef cattle.

In broilers, the ADG was improved when glycyrrhiza polysaccharides were supplemented in the diet [22]. You et al. reported that adding licorice flavonoids to the diet of weaned piglets tended to increase ADG and reduce F/G [23]. In addition, LE enhanced the growth performance of fattening lambs [4]. The current data revealed that the dietary addition of LE significantly increased the ADG and reduced the F/G, improving the growth performance of beef cattle in the middle of the experiment. Meanwhile, the nutrient digestibility of CP, CF, and Ca was enhanced when the diets were supplied with the LE of cattle. Therefore, we affirmed that adding LE to the diet could enhance the absorption of nutrient substances in beef cattle, thereby improving the growth performance of beef cattle in the middle of the experiment. As for the growth performance of the whole experiment period, which was not significant, it may be due to the high temperature in the later period, which affected the growth of beef cattle; but, the specific mechanism needs to be further explored [24].

A previous study showed that ALP activity was positively correlated with animal production performance [25]. The increased ALP levels in the serum of the current study uncovered that LE could improve the production performance of beef cattle, which was similar to a previous study on calves [18]. This was similar to another study, which manifested that licorice flavonoid powder could enhance the activity of ALP in the blood of weaned piglets [19]. At the same time, it was discovered that a significant reduction in CHOL in the study occurred, which was similar to the research on broilers by Naser et al. [26]. The decreased CHOL might be caused by the easy absorption of the intestine from phytosteroids in licorice [27]. Alternatively, the saponins in licorice may also form insoluble substances with cholesterol, which may lead to the obstruction of intestinal cholesterol absorption [28]. But, the specific mechanism requires further investigation.

In our investigation, adding LE to the beef cattle diet considerably increased the level of INS, implying that LE could boost sugar conversion and consumption in beef cattle [29]. Furthermore, whereas cattle receiving LE had no significant effect on GH content, which ascended in the CHM group compared with that in the CK group. The low amount of GH in plasma was related to the poorer growth rates in porcine [30]. These findings suggested that the addition of LE in the diet may promote the transformation and absorption of some substances, to some extent, and accelerate the growth rate of beef cattle.

IgM and IgG were the major mediators of humoral immunity. IgA bolstered the local mucosal defense against infection of the body despite its content being low in the blood [31,32,33]. The serum IgA content of beef cattle receiving LE supplementation was increased and this was consistent with a previous study’s results that IgA in the serum was raised after LE supplementation in Karakul Sheep, which indicated that LE could enhance the local mucosal immune function of the body [34]. Li et al. showed that adding glycyrrhiza polysaccharide to the basal diet of weaned piglets could increase the content of IgA and IgG in their serum [35]. Sajjadi et al. reported that female Holstein calves receiving licorice addition had greater IgA and IgG levels in serum [36]. These results suggested that LE had a greater improvement in the immunity of animals.

GSH-Px, CAT, and T-AOC were the main antioxidant indexes of the body. MDA was an indicator of lipid oxidation [37,38]. Our findings indicated that feeding LE substantially enhanced the CAT and NO content in serum but the contents of GSH Px, MDA, and T-AOC were similar compared with the CK group. A prior study reported that weaned Piglets receiving LE addition had no discernible impact on CAT content but the content of GSH-Px was greatly improved [39]. Jiang et al. revealed that adding licorice extract to the diet of weaned yaks could increase the concentration of GSH-Px and decrease the content of MDA in the serum of calves [18]. Zhang et al. showed that dietary glycyrrhiza polysaccharides could enhance serum GSH and decrease the MDA content in broilers [7]. The inconsistent results were possibly due to differences in the subjects studied, the duration of the experiment, and the structure and content of specific active ingredients in LE.

We further investigated the effects of LE on the hindgut fecal microbiota of beef cattle, which may be a factor that improved the growth performance of beef cattle. Our study confirmed that no significant differences were found in the Alpha Diversity Analysis and that the two groups overlapped in the PCoA chart of Beta diversity analysis, which is consistent with the study of Zhu et al. [39]. This indicated that after adding licorice extract to the beef cattle diet, the fecal microbiota was still dominated by the core community and the addition of licorice extract did not affect the stability of the fecal bacterial community structure [40]. Firmicutes and Bacteroidota accounted for the largest proportion of hindgut fecal microbiota, which was consistent with previous reports on bacterial communities [22,41]. As the dominant phyla, Firmicutes played a role in carbohydrate metabolism, producing short-chain fatty acids and inhibiting inflammation [42]. Bacteroidota promoted the absorption and utilization of polysaccharides [43]. The nutrient absorption was promoted when the abundance of Firmicutes was higher than that of Bacteroidota [44]. Based on the aforementioned relevant research and analysis, we speculated that LE could promote the digestion of nutrients by changing the composition of the flora, thus improving the growth performance of beef cattle in the middle of the experiment.

Our results suggested that adding LE to the beef cattle diet could significantly increase the abundance of Ileibacterium, p-2534-18b5-gut-group, and [Eubacterium]-oxidoreducens-group and significantly reduce the relative abundance of Gallinobacterium and Breznakia. SCFAs were produced by Ileibacterium to adjust the intestinal health of animals. Breznakia was a fermenting bacterium isolated from the gut of insects [45,46,47]. [Eubacterium]-oxidoreducens-group, a bacterium of intestinal butyricogenes, by which the butyric acid was produced to enhance the intestinal barrier function of animals, improve intestinal health, and increase feed digestibility [48,49]. Gallibacterium belongs to the Pasteurella family, which is closely related to salpingitis, peritonitis, and ovaritis in poultry and has the potential to cause disease [50]. Ma et al. found that the abundance of p-2534-18b5-gut-group in the gut of stunted yaks was significantly lower than normal growing yaks [51]. A previous study showed that adding glycyrrhiza polysaccharides to broiler diets promoted the secretion activity of cuprocytes, altered the diversity and abundance of gut microbes, increased the relative abundance of beneficial bacteria (Bacteroides, Butyricicoccus, Ruminococcaceae, and Lactobacillus), and decreased the relative abundance of harmful bacteria to the host (Erysipelatoclostridium, Lachnoclostridium, and Escherichia-Shigella) [21]. The aforementioned research discovered that LE could encourage the healthy growth of beef cattle by reducing the number of harmful bacteria and increasing the number of beneficial bacteria in the hindgut of beef cattle.

We realized that the addition of LE in the beef cattle diet increased the levels of corticosterone and that cortodoxone belonged to the metabolic pathways of steroid hormone biosynthesis and reduced the levels of biotin and 2-phenylethylamine, which were subordinate to biotin metabolism and phenylalanine metabolism pathways. Steroid hormones could promote protein synthesis and muscle development and stimulate osteoblast proliferation and differentiation, thereby accelerating the body’s development [52]. Biotin is an essential nutrient factor for animal growth, development, and reproduction. The accumulation of phenylalanine could promote inflammatory reactions in animals [53,54]. Feng et al. found that adding phenylalanine to the grass carp diet can reduce the mRNA levels of SOD, IL-8, TNF-α, and nuclear factor NF-κB p65 in the gut [55]. At the same time, the correlation analysis disclosed that the upregulation of p-2534-18b5-gut-group caused the increase in metabolites of cortodoxone and corticosterone. The decrease in 2-phenylethylamine content caused by the down-regulation of Gallibacterium could inhibit phenylalanine metabolism. These results suggest that LE could promote beef cattle growth and prevent diseases by improving fecal microbial community and metabolic differences, which is beneficial to the health of beef cattle. However, the specific mechanisms of how LE changed post-intestinal fecal metabolite differences by affecting the fecal microbial community need to be further studied.

## 5. Conclusions

In this study, the supplementation of LE promoted growth performance, blood biochemical indices, and the health status of beef cattle in the middle of the experiment. Specifically, the supplementation of LE in the beef cattle diet could improve the ADG and the digestibility of CP, CF, and Ca and reduce the F/G. LE supplementation could regulate the levels of ALB, ALP, CHOL, INS, IgA, CAT, and NO in beef cattle serum to improve and enhance the material metabolism, antioxidant, and immune performance of beef cattle. In terms of hindgut feces, LE could significantly enhance the abundance of Ileibacterium, p-2534-18b5-gut-group, and [Eubacterium]-oxidoreducens-group and decrease the abundance of Gallibacterium and Breznakia to regulate steroid hormone biosynthesis and phenylalanine metabolism pathways to promote the healthy growth of beef cattle.

## Figures and Tables

**Figure 1 vetsci-11-00356-f001:**
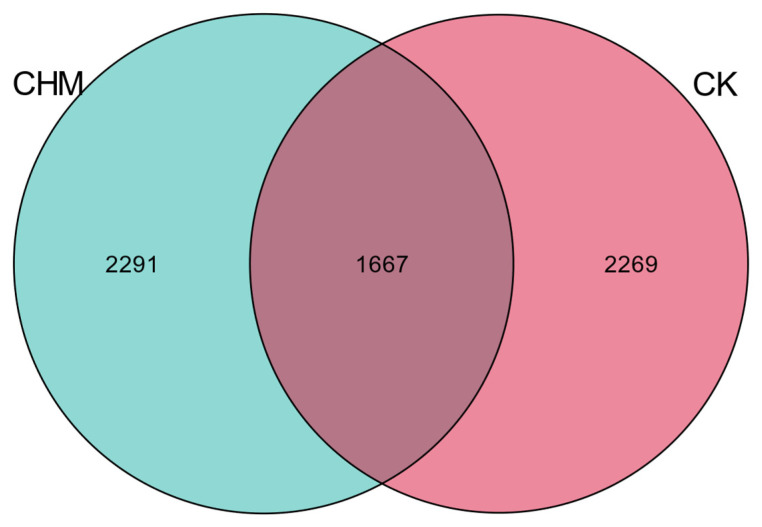
OUT Venn diagram of fecal microbes in the hindgut. Each circle in the figure represents a group. The numbers without overlapping areas represent the unique OTUs of each group. The CK and the CHM, respectively, represent the control group and the experimental group.

**Figure 2 vetsci-11-00356-f002:**
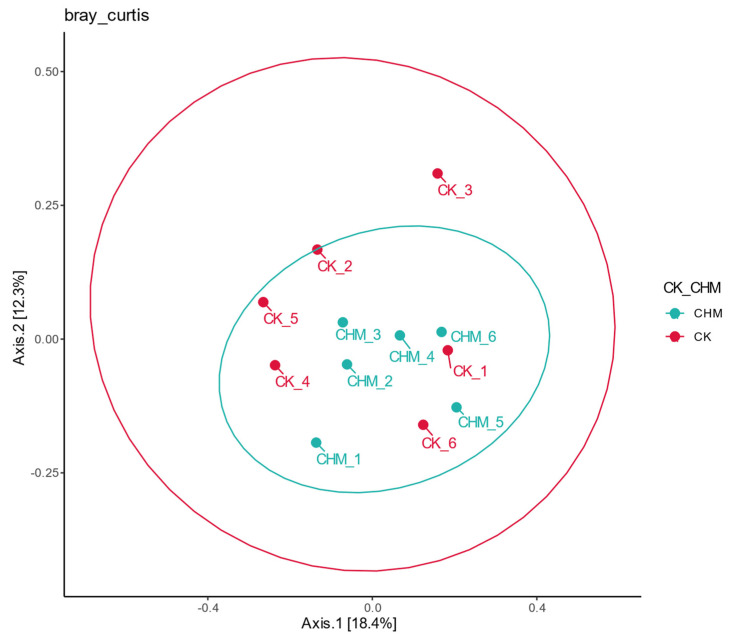
Microbial sequences of beef cattle fecal samples were subjected to PCoA analysis for Beta diversity analysis. On the 2D coordinate, Axis 1 and Axis 2 accounted for 18.4% and 12.3% of the interpretation of the results, respectively. The CK and the CHM, respectively, represent the control group and the experimental group.

**Figure 3 vetsci-11-00356-f003:**
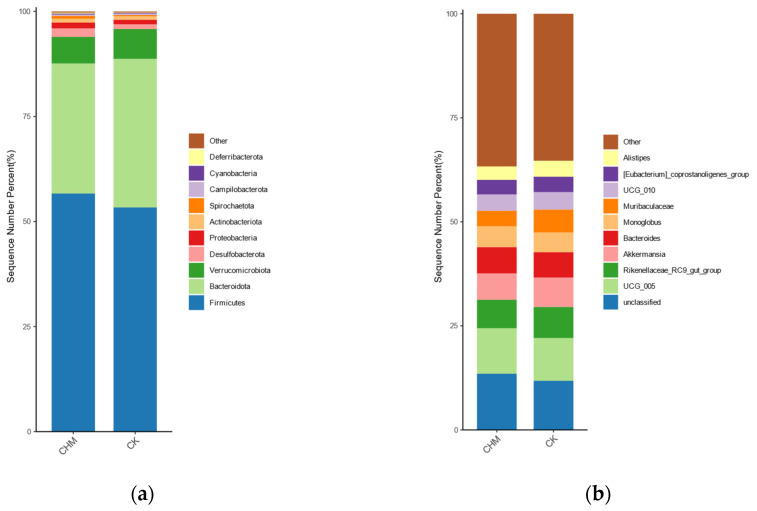
Differences in the fecal bacterial taxa in beef cattle between the CK and the CHM. (**a**) Phyla level taxa. (**b**) Genus level taxa. The CK and the CHM, respectively, represent the control group and the experimental group.

**Figure 4 vetsci-11-00356-f004:**
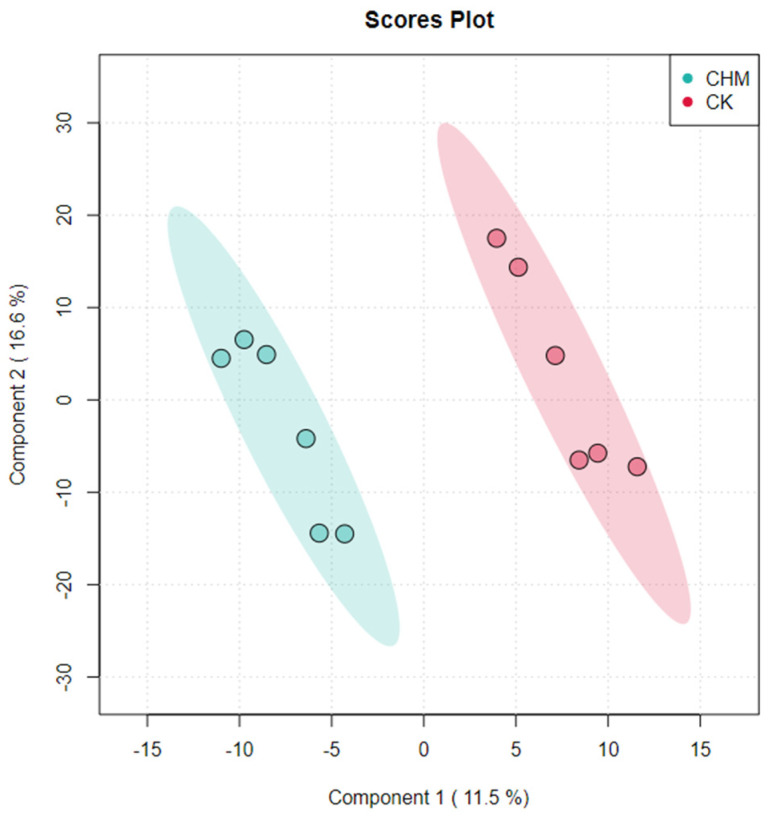
Least squares discriminant analysis (PLS−DA) score plot of fecal metabolic differences between the experimental group and the control group cattle. Hindgut fecal metabolites were separated and clustered in different regions. The CK and the CHM, respectively, represent the control group and the experimental group.

**Figure 5 vetsci-11-00356-f005:**
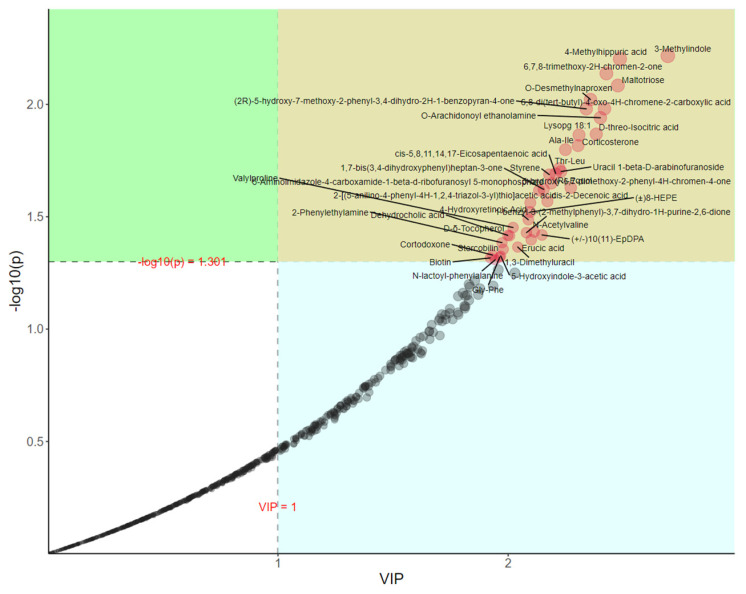
Importance map of PLS−DA metabolites in the fecal metabolome of beef cattle. The threshold was set to vip > 1 and *p* < 0.05, obtaining 39 different metabolites. VIP = variable importance in the projection.

**Figure 6 vetsci-11-00356-f006:**
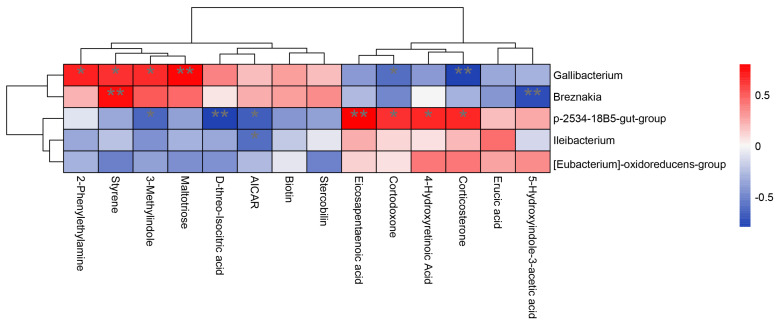
Numerical heat map for the correlation analysis of differential metabolites and fecal genus−level differential flora. Red means positive correlation and blue means the opposite. The darker the color, the more relevant it is. * means significant correlation and ** means highly significant correlation.

**Table 1 vetsci-11-00356-t001:** The influence of licorice extract on the growth and nutrient digestibility of beef cattle.

Item	Day	Group	
		CK	CHM
Initial weight, kg		221.67 ± 9.37	215.83 ± 8.11
Final weight, kg		270.83 ± 9.44	270.83 ± 8.70
ADG, kg/d	0–30 d	0.56 ± 0.06	0.47 ± 0.05
	30–60 d	0.53 ± 0.03 ^B^	0.78 ± 0.07 ^A^
	60–90 d	0.56 ± 0.04	0.59 ± 0.04
	0–90 d	0.55 ± 0.03	0.61 ± 0.03
ADFI, kg/d	0–30 d	7.45 ± 0.38	7.68 ± 0.23
	30–60 d	7.28 ± 0.29	7.07 ± 0.23
	60–90 d	7.61 ± 0.31	7.58 ± 0.25
	0–90 d	7.45 ± 0.32	7.44 ± 0.23
F/G, kg/kg	0–30 d	14.33 ± 1.95	17.40 ± 2.18
	30–60 d	13.91 ± 0.74 ^A^	9.45 ± 0.86 ^B^
	60–90 d	14.02 ±1.11	13.28 ± 0.99
	0–90 d	13.81 ± 0.89	12.31 ± 0.67
CP, %	30 d	71.58 ± 0.66	69.60 ± 0.79
	60 d	66.95 ± 1.50 ^b^	71.41 ± 1.01 ^a^
	90 d	77.81 ± 1.27	80.02 ± 0.83
CF, %	30 d	38.06 ± 2.97 ^b^	51.41 ± 3.54 ^a^
	60 d	28.01 ± 3.16	28.33 ± 2.92
	90 d	33.92 ± 2.42	38.49 ± 0.67
EE, %	30 d	67.16 ± 1.87	69.20 ± 0.69
	60 d	58.05 ± 2.30	65.54 ± 2.45
	90 d	76.45 ± 2.08	75.38 ± 1.25
Ash, %	30 d	24.44 ± 1.19	24.45 ± 1.32
	60 d	24.36 ± 1.03	24.57 ± 1.12
	90 d	35.37 ± 1.12	37.69 ± 1.14
ADF, %	30 d	53.36 ± 1.35	52.39 ± 3.91
	60 d	37.67 ± 2.19	39.32 ± 1.82
	90 d	56.00 ± 1.75	56.95 ± 0.33
NDF, %	30 d	51.32 ± 1.32	48.84 ± 1.83
	60 d	41.86 ± 1.59	42.77 ± 0.52
	90 d	56.50 ± 1.11	57.17 ± 0.14
GE, %	30 d	61.26 ± 1.30	57.65 ± 1.17
	60 d	54.05 ± 3.63	57.64 ± 0.90
	90 d	67.96 ± 1.03	68.12 ± 0.86
Ca, %	30 d	36.96 ± 2.05 ^b^	45.76 ± 1.69 ^a^
	60 d	46.29 ± 3.68	46.14 ± 2.05
	90 d	54.62 ± 1.78	56.55 ± 1.01
P, %	30 d	30.33 ± 1.03	32.30 ± 1.32
	60 d	23.40 ± 2.10	21.25 ± 2.18
	90 d	48.82 ± 3.34	43.06 ± 5.30

ADG = average daily gain; ADFI = average dry feed intake; F/G = feed to meat ratio; CP = crude protein; CF = crude fiber; EE = crude fat; crudeAsh = Ash; ADF = acid detergent fiber; NDF = neutral detergent fiber; GE = general energy. CK = control group; CHM = Licorice extract group. ^A,B^ Different letters in a row mean extremely significant differences (*p* < 0.01). ^a,b^ Different letters in a row mean significant differences (*p* < 0.05).

**Table 2 vetsci-11-00356-t002:** The influence of licorice extract on blood biochemical indexes and hormones of beef cattle.

Item	Day	Group	
		CK	CHM
ALB, g/L	30 d	17.15 ± 0.54	17.00 ± 0.61
	60 d	18.43 ± 0.37	18.50 ± 0.49
	90 d	15.73 ± 0.56 ^b^	18.48 ± 0.66 ^a^
ALP, U/L	30 d	125.50 ± 14.64	109.00 ± 4.81
	60 d	114.00 ± 10.12 ^b^	163.00 ± 12.25 ^a^
	90 d	172.25 ± 23.58	187.00 ± 16.61
ALT, U/L	30 d	18.75 ± 0.63	19.75 ± 0.48
	60 d	19.00 ± 1.87	17.75 ± 0.85
	90 d	21.50 ± 5.01	18.75 ± 0.85
AST, U/L	30 d	38.00 ± 1.41	39.00 ± 1.68
	60 d	35.25 ± 2.66	34.50 ± 1.32
	90 d	45.00 ± 4.56	44.00 ± 2.68
CHOL, mmol/L	30 d	2.06 ± 0.27	2.15 ± 0.17
	60 d	2.55 ± 0.09 ^b^	2.08 ± 0.14 ^a^
	90 d	2.42 ± 0.27	2.60 ± 0.11
TG, mmol/L	30 d	0.10 ± 0.00	0.10 ± 0.00
	60 d	0.10 ± 0.00	0.10 ± 0.00
	90 d	0.13 ± 0.03	0.13 ± 0.03
TP, g/L	30 d	35.25 ± 0.87	40.43 ± 1.97
	60 d	37.20 ± 1.77	38.45 ± 1.44
	90 d	31.05 ± 1.89	35.43 ± 1.36
Urea, mmol/L	30 d	3.65 ± 0.17	4.28 ± 0.21
	60 d	3.85 ± 0.12	4.23 ± 0.18
	90 d	3.58 ± 0.13	3.68 ± 0.08
INS, μIU/mL	30 d	10.13 ± 0.65 ^b^	16.26 ± 1.54 ^a^
	60 d	11.19 ± 0.89	12.24 ± 0.51
	90 d	9.37 ± 0.75	10.72 ± 0.51
GH, ng/mL	30 d	0.64 ± 0.12	1.06 ± 0.12
	60 d	0.74 ± 0.02	0.76 ± 0.03
	90 d	0.69 ± 0.03	0.83 ± 0.15

AST = aspartate aminotransferase; ALP = alkaline phosphatase; TP = total protein; ALB = albumin; UREA = urea; TG = triglyceride; CHOL = cholesterol; ALT = alanine aminotransferase. CK = control group; INS = insulin; GH = growth hormone. CHM = Licorice extract group. ^a,b^ Different letters in a row mean significant differences (*p* < 0.05).

**Table 3 vetsci-11-00356-t003:** The influence of licorice extract on the immune performance of beef cattle.

Item	Day	Group	
		CK	CHM
IgA, g/L	30 d	0.50 ± 0.00	0.50 ± 0.04
	60 d	0.50 ± 0.06	0.50 ± 0.04
	90 d	0.45 ± 0.03 ^b^	0.50 ± 0.04 ^a^
IgG, g/L	30 d	0.33 ± 0.05	0.30 ± 0.07
	60 d	0.35 ± 0.09	0.30 ± 0.07
	90 d	0.33 ± 0.08	0.35 ± 0.05
IgM, g/L	30 d	0.13 ± 0.03	0.15 ± 0.03
	60 d	0.15 ± 0.06	0.15 ± 0.06
	90 d	0.13 ± 0.03	0.13 ± 0.03

CK = control group; CHM = Licorice extract group. ^a,b^ Different letters in a row mean significant differences. (*p* < 0.05).

**Table 4 vetsci-11-00356-t004:** The influence of licorice extract on the antioxidant function of beef cattle.

Item	Day	Group	
		CK	CHM
GSH-Px, U/mL	30 d	66.08 ± 11.11	57.85 ± 3.68
	60 d	112.61 ± 5.35	107.38 ± 1.18
	90 d	65.49 ± 9.05	61.96 ± 6.11
CAT, U/mL	30 d	1.52 ± 0.27	3.00 ± 0.58
	60 d	2.76 ± 0.06	2.37 ± 0.17
	90 d	1.51 ± 0.14 ^b^	2.15 ± 0.13 ^a^
MDA, nmol/L	30 d	1.44 ± 0.04	1.61 ± 0.08
	60 d	2.04 ± 0.11	2.03 ± 0.02
	90 d	1.70 ± 0.15	2.18 ± 0.19
NO, μmol/L	30 d	30.06 ± 1.15 ^b^	40.34 ± 2.64 ^a^
	60 d	31.78 ± 1.28	35.43 ± 2.39
	90 d	38.06 ± 0.99	39.60 ± 0.81
T-AOC, mM	30 d	2.05 ± 0.03	2.09 ± 0.03
	60 d	2.22 ± 0.04	2.26 ± 0.05
	90 d	2.28 ± 0.09	2.36 ± 0.05

CAT = catalase; GSH-Px = glutathione peroxidase; T-AOC = total antioxidant energy; NO = nitric oxide; MDA = malondialdehyde. CK = control group; CHM = Licorice extract group. ^a,b^ Different letters in the same line mean significant differences (*p* < 0.05).

**Table 5 vetsci-11-00356-t005:** The influence of licorice extract on level strains of the fecal bacterial community of beef cattle.

Bacterial Genu	Group	
	CK	CHM
Gallibacterium, %	0.0001 ^a^	0 ^b^
Ileibacterium, %	0.0012 ^b^	0.0061 ^a^
p-2534-18b5-gut-group, %	0.0009 ^b^	0.0021 ^a^
[Eubacterium]-oxidoreducens-group, %	0.0003 ^b^	0.0006 ^a^
Breznakia, %	0.0001 ^a^	0 ^b^

CK = control group; CHM = Licorice extract group. ^a,b^ Different letters in a row mean significant differences (*p* < 0.05).

**Table 6 vetsci-11-00356-t006:** Metabolic differences between the CK and the CHM groups of beef cattle posterior intestinal feces metabolites (VIP > 1 and *p* < 0.05).

Category	Metabolite	CK	CHM	VIP	FDR	*p*-Value
Lipids and lipid-like molecules	4-hydroxyretinoic acid	0.858	1.935	2.09	0.077	0.033
	Eicosapentaenoic acid	2.133	3.552	2.21	0.077	0.020
	Corticosterone	0.099	0.198	2.30	0.077	0.015
	Cortodoxone	2.574	3.572	1.93	0.058	0.047
	Erucic acid	0.522	0.703	2.10	0.058	0.040
Organic heterocyclic compounds	3-methylindole	0.545	0.144	2.69	0.077	0.006
	5-hydroxyindole-3-acetic acid	0.935	2.229	1.97	0.064	0.047
	Biotin	2.359	2.019	1.92	0.058	0.048
	Stercobilin	842.258	635.457	1.98	0.058	0.044
Benzenoids	2-phenylethylamine	0.510	0.200	1.97	0.058	0.041
	Styrene	18.418	11.198	2.18	0.077	0.021
Organic acids and derivatives	D-threo-Isocitric acid	0.405	0.277	2.38	0.077	0.014
Nucleosides, nucleotides, and analogues	AICAR	0.446	0.362	2.13	0.077	0.024
Organic oxygen compounds	Maltotriose	0.252	0.153	2.48	0.077	0.008

VIP = variable importance in the projection. CK = control group; CHM = Licorice extract group.

**Table 7 vetsci-11-00356-t007:** Pathway analysis of hindgut fecal metabolites between the CK and the CHM groups in beef cattle.

Metabolic Pathway	Total Metabolite	Number of Differential Metabolite	Metabolite	*p*-Value	Impact	State of Affair
Steroid hormone biosynthesis	85	2	Corticosterone and Cortodoxone	0.017	0.0303	Up regulation
Phenylalanine metabolism	10	1	2-Phenylethyla-mine	0.026	0.1111	Up regulation
Biotin metabolism	10	1	Biotin	0.026	0.1250	Down regulation

## Data Availability

Data can be made available via email to Shuilian Wang, wangshuilian1234@126.com.

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
