# Peer review of "Licorice Extract Supplementation Benefits Growth Performance, Blood Biochemistry and Hormones, Immune Antioxidant Status, Hindgut Fecal Microbial Community, and Metabolism in Beef Cattle"

_vetsci, 2024, doi:10.3390/vetsci11080356_

Round 1
Reviewer 1 Report
Comments and Suggestions for Authors
This is a nice paper with good work. Unfortunately, there is significant grammatical mistakes, particularly with present and past tense use. My original review was not saved for some reason and I am going to share an abbreviated synopsis of the review as all of my line by line suggestions were not entered into the system.
The experimental design is unclear. Were the animals pen fed or individually housed?
You have very minor differences with only small differences in 1 period for the variables that were statistically different. In the overall, performance which is most important, no differences were observed. Don't over exaggerate minor differences when using small N.
The discussion is superficial and requires significant expansion. You must relate your findings with previous research. Did the cattle perform as well? Were you blood metabolites similar? When I reviewed your metabolites, they were not in line with standard reference values which makes me question your lab results. Further, did you validate the lab kits used to ensure they would work with the animals and sample type used? How do you know what your recovery rates were for you samples? Point is that I think you will find that your results do not align with other bovine data.
Your charts and figures need to stand alone and be legible.
This is a novel area of work and would add to the body of information, but requires major edits for further consideration.
Comments on the Quality of English LanguageNeeds significant grammatical work
Reviewer 2 Report
Comments and Suggestions for Authors
This study characterized that the addition of licorice extract to the beef cattle diet was able to benefit the digestibility of some nutrients, serum parameters (biochemical, hormonal, immune and antioxidant), hindgut fecal microbiota and metabolome of beef cattle, eventually to promote growth performance and maintain the health of animals. The article describes the experiment in detail, including the objectives, methods and conclusions. The experiment was well designed and the study is meaningful. Adequate workload supports the article content well. Below are some notes that may contribute to improving the manuscript.
1. please provide the codes and specifications of the licorice extract
2. Line 24-25: the entire experimental phase lasted for 90 days,but in line 80 The trial period was 120 days. This is confusing.
3. Line 109: Please specifically describe the number of times and weight of dung collected from beef cattle.
4. Line 168: change “30, greater CP digestibility (p < 0.05) ” to “30 and greater CP digestibility (p < 0.05)”. please check the full text.
5. Line 169: Explain the meaning of CK and CHM in the table. please check the full text.
6. Line 195: change “ml” to “mL”. please check the full text.
7. Line 207-Line209: change “The numbers of circles and overlapping areas represent the total number of OTUs between the groups” to “The number of overlapping area represents the total number of OTUs shared between the groups”. please check the full text
8. Line 283-285: Has licorice extract been studied on animals other than those mentioned in this article? If so, it should be mentioned.
9. Please, check the statistical analysis model as for some parameters, like DMI, does not fit (repeated measurements). Line 181: which kind of test did the authors choose?
10. The tables in the manuscript are too many, try to merge or put some in the attachment.
Comments on the Quality of English LanguageModerate editing of English language required
Reviewer 3 Report
Comments and Suggestions for Authors
I ask you to comment on what was the basis for deciding to include only 2 g of Licorice extract and not more or less quantities. I do not see a comment that supports this decision.
The statistical analysis was an ANOVA, it would be advisable to apply at least another analysis such as Studente's "t" since there were only two treatments.
I suggest a brief comment from you if you consider that only six repetitions per treatment (animals) provide sufficient degrees of freedom for experimental error and to detect differences between the means of results.
I ask you to comment on what was the basis for deciding to include only 2 g of Licorice extract and not more or less quantities. I do not see a comment that supports this decision.
The statistical analysis was an ANOVA, it would be advisable to apply at least another analysis such as Studente's "t" since there were only two treatments.
I suggest a brief comment from you if you consider that only six repetitions per treatment (animals) provide sufficient degrees of freedom for experimental error and to detect differences between the means of results.
Author Response
Please see the attachmeng.
